# Determining vitreous viscosity using fluorescence recovery after photobleaching

Nishanthan Srikantha[1,2]*, Yurema Teijeiro-Gonzalez[3], Andrew Simpson[1,2], Naba Elsaid[4], Satyanarayana Somavarapu[5], Klaus Suhling[3], Timothy L. Jackson[1,2]

**1** School of Medicine, King's College London, London, United Kingdom, **2** Department of Ophthalmology, King's College Hospital, London, United Kingdom, **3** Department of Physics, King's College London, Strand, London, United Kingdom, **4** Anglia Ruskin University, Bishop Hall Lane, Chelmsford, United Kingdom, **5** Department of Pharmaceutics, University College London School of Pharmacy, London, United Kingdom

* nish.srikantha@nhs.net

**Data Availability Statement:** All relevant data has been included as table and graph format in the original paper and can be easily read using online tools as standard.

## Abstract

### Purpose

Vitreous humor is a complex biofluid whose composition determines its structure and function. Vitreous viscosity will affect the delivery, distribution, and half-life of intraocular drugs, and key physiological molecules. The central pig vitreous is thought to closely match human vitreous viscosity. Diffusion is inversely related to viscosity, and diffusion is of fundamental importance for all biochemical reactions. Fluorescence Recovery After Photobleaching (FRAP) may provide a novel means of measuring intravitreal diffusion that could be applied to drugs and physiological macromolecules. It would also provide information about vitreous viscosity, which is relevant to drug elimination, and delivery.

### Methods

Vitreous viscosity and intravitreal macromolecular diffusion of fluorescently labelled macromolecules were investigated in porcine eyes using fluorescence recovery after photobleaching (FRAP). Fluorescein isothiocyanate conjugated (FITC) dextrans and ficolls of varying molecular weights (MWs), and FITC-bovine serum albumin (BSA) were employed using FRAP bleach areas of different diameters.

### Results

The mean (±standard deviation) viscosity of porcine vitreous using dextran, ficoll and BSA were 3.54 ± 1.40, 2.86 ± 1.13 and 4.54 ± 0.13 cP respectively, with an average of 3.65 ± 0.60 cP.

### Conclusions

FRAP is a feasible and practical optical method to quantify the diffusion of macromolecules through vitreous.

**Funding:** The project was funded by Fight for Sight Grant Number 1874 and BBSRC Grant Number BB/R004803/1. The funders had no role in study design, data collection and analysis, decision to publish, or preparation of the manuscript. N. Srikantha is a Fight for Sight Research Fellow. K. Suhling acknowledges BBSRC grant BB/R004803/1.

**Competing interests:** The authors have declared that no competing interests exist.

## Introduction

Many posterior segment diseases are treated with the administration of intravitreal therapeutics. These agents move through the vitreous to cause an effect at their designated target sites, such as the macula. The vitreous humor is a complex biofluid located between the lens and retina in the eye. It is composed of collagen, hyaluronic acid (HA) and proteoglycans but is >99% water [1]. As a complex biofluid, the vitreous exhibits both liquid and solid behaviour [2, 3] and its rheological properties are given by the presence and distribution of collagen and HA [3, 4]. Water is trapped by HA molecules interspersed between linearly arranged collagen fibres. The organized structure of the interwoven collagen and HA creates a diffusion barrier for cells and macromolecules, but small molecules such as water and electrolytes diffuse more freely [5]. The total human vitreous volume is approximately 4 mL, and the macromolecular structure of vitreous varies with age. It exists as a firm gel in youth but becomes syneretic with age [6]. By age 70, vitreous liquefaction and collapse means that 50% of eyes have vitreous detachment from the posterior retina [7, 8]. Researchers modelling drug movement within the vitreous postulate that the retrozonular space of petit, a gap between the anterior surface of the vitreous and the ciliary body, is an important channel for drug movement [9]. Different regions of the vitreous have different viscosity [2, 4, 10, 11]. The viscosity is highest in the posterior vitreous and decreases towards the anterior segment [11–13].

Viscosity is the resistance of a fluid to flow and is a key property affecting diffusion. Viscosity can be measured by mechanical methods, such as capillary or rotational viscometers or rheometers [2, 3, 6] but optical methods based on fluorescent dyes dissolved in the fluid can also be employed. Viscosity values derived from mechanical methods depend on the conditions applied to measure them, for example Silva et al [3] report a drop of the shear viscosity of vitreous humor as a function of shear stress from around 200,000 cP below a shear stress of 0.2 Pa to around 3 cP above this value [3].

Magnetic resonance imaging (MRI) has also been employed not only to measure, but also to image the rheological properties of vitreous, both in porcine [14] and human eyes (post mortem) [15]. The so-called transverse relaxation time T2 is correlated to viscoelastic properties of the vitreous, i.e. the storage or loss moduli (which are related to the damping properties of the material) and the viscosity. The great advantage of this approach is that it maps T2 and thus provides images which allows spatial variation of viscoelastic properties to be visualised. For example, the central vitreous has a slightly shorter T2 compared to the peripheral vitreous [14, 15].

To assess diffusion and viscosity as it is experienced of drug molecules, we employed fluorescently labelled macromolecules, of similar molecular weight to eye drugs, and used fluorescence recovery after photobleaching (FRAP), an optical method to measure viscosity and diffusion. This approach has further advantages, e.g. only very small sample volumes are needed, and viscosity measurements can be carried out via fluorescence microscopy on the scale of micrometers.

Information regarding viscosity of vitreous is important for many reasons. Differences in viscosity throughout the vitreous body could guide potential injection sites within the vitreous when administering intravitreal therapeutic agents. The differences in diffusion between saline and vitreous is directly relevant to the pharmacokinetics of drugs injected into the vitreous following pars plana vitrectomy [16–18], and perhaps also following intravitreal ocriplasmin (ocriplasmin is used to dissolve the vitreous as part of a so-called chemical vitrectomy) [19], and in ageing syneretic vitreous. Drugs such as ranibizumab (48 kDa) are similar in molecular weight (MW), and bevacizumab (150 kDa] and aflibercept (115 kDa) are less than double the MW [20–24] of the fluorescently labelled macromolecules studied in this work, and their

diffusion properties are important for treating age-related macular degeneration. Moreover, polymeric hydrogels have been investigated as vitreous substitutes [4, 25]. In addition to biocompatibility and functionality, the optical and rheological properties of these substitute materials, e.g., refractive index, density, water content, stiffness, viscoelasticity and also viscosity, should be as close as possible to the original vitreous humor.

We previously used FRAP to study the diffusion of macromolecular ophthalmic drugs [26]. Since it has become relatively easy to label these large drugs with fluorescent dyes, FRAP provides an opportunity to study macromolecular drug mobility in solution, as well as across biological tissue and cells.

Here, we use FRAP to study intravitreal diffusion to provide information about the viscosity of the vitreous humor, a parameter which is relevant to drug elimination, and delivery.

## Materials and methods

### Sample preparation

Porcine eyes were purchased from a local abattoir. Eyes were removed and supplied to the laboratory within 24 hours. Eyes were stored in aqueous solution in a refrigerator at 5°C for no more than 24 hours prior to vitreous dissection. Information regarding age and sex of the eyes are unavailable. The extra-ocular muscles were dissected free and the anterior segment was removed by cutting circumferentially behind the limbus. Vitreous was carefully dissected away using fine forceps and spring scissors, keeping the vitreous whole throughout the dissection process. The vitreous body was then dissected in half, and the central region with a diameter less than 1 cm was carefully removed from both halves. We used three eyes altogether. From two eyes, we prepared three vitreous body samples for each. These three separate central vitreous body samples of around 1 mL volume each were soaked for 24 hours in about 5 mL of $10^{-5}$ molar fluorescein-isothiocyanate (FITC) labelled, neutral dextran or ficoll, each of either 20, 40 and 70 kDa (TdB Labs, Uppsala, Sweden). According to the manufacturer, the fluorescein moiety is attached by a stable thiocarbamoyl linkage and the labelling procedure does not lead to any depolymerization of the dextran. FITC-dextrans have from 0.002–0.008 mol FITC per glucose unit. Ficoll (polysucrose) is a polymer synthesised by cross-linking sucrose with epichlorohydrin. FITC-ficoll is prepared by reacting ficoll with FITC under similar conditions to those used for FITC-dextrans. Ficoll has a more spherical structure and is less flexible compared to dextran. The central vitreous from the third eye was soaked in neutral 66.5 kDa FITC-bovine serum albumin (BSA) (Sigma, Poole, UK).

### Fluorescence Recovery After Photobleaching (FRAP)

FRAP is an optical technique capable of providing quantitative two-dimensional lateral diffusion analysis of samples containing fluorescently labelled probes. The method employs the irradiation (bleaching) of a fluorophore in the focal area of a light beam, usually laser light [27]. This abolishes fluorescence and then a highly attenuated light beam measures the recovery of the fluorescence in the bleach area. The fluorophore is usually permanently bleached in the area of the focal beam; however, fluorescence returns to the bleached area due to diffusion of unbleached fluorophore from the surrounding area [28]. This recovery is a measure of the diffusion of the fluorophore.

200 μL vitreous, containing the FITC-labelled molecules, was placed into one well of a black, glass bottom, 96-well micro well plate (Greiner, Germany). FRAP was performed using three different bleach areas, using a modification of techniques described previously [26]. A confocal laser scanning microscope (Leica TCS SP2) was used to photobleach a circular area (with an excitation wavelength $\lambda_{ex}$ = 488 nm) containing the FITC-labelled molecules ($\lambda_{em}$ =

510 nm). The photobleaching was performed in the middle of the vitreous sample using measured z-axis readings as a guide. This was done to avoid experiments being performed on FITC solution that may have accumulated on the outer aspect of the vitreous. The speed with which neighbouring molecules diffused back into this region of interest was determined by measuring the fluorescence recovery $F(t)$ using a bi-exponential fit.

$$F(t) = a - be^{-ct} - de^{-ft} \qquad (1)$$

where $a$, $b$, $c$, $d$ and $f$ are constants. $a$ is the pre-bleach fluorescence average of the bleach area, $b$ and $d$ are pre-exponential factors and $c$ and $f$ are recovery rates. Bleach areas with a radius of 15 μm, 20 μm and 25 μm were used, and the experiments were performed at room temperature.

## Theoretical framework and data analysis

The Stokes–Debye–Einstein equation can be used to calculate the diffusion coefficient $D$ of a spherical particle of radius $r$ undergoing Brownian motion in a stable fluid at uniform temperature [29]. The equations below separate the formula into stages to clearly demonstrate how the calculations were performed:

$$D = \frac{kT}{6\pi\eta r} \qquad (2)$$

where $k$ is the Boltzmann constant, $T$ the temperature in Kelvin, and $\eta$ the viscosity.

The translational diffusion coefficient ($D$) is proportional to the half-recovery time $\tau_{1/2}$ according to the equation [28],

$$D = \gamma\frac{w^2}{4\tau_{1/2}} \qquad (3)$$

where $\gamma$ is 0.88 and w is the bleach radius. The diffusion coefficient has dimensions of $[\text{length}]^2/[\text{time}] = [\text{area}]/[\text{time}]$.

Assuming a hard sphere model for molecules, the relationship between the radius and the molecular weight is:

$$r = 0.67(MW)^{1/3} \qquad (4)$$

where MW is the molecular weight and 0.67 is expressed in ÅDa$^{-1/3}$ [30]. The radius for the 20, 40 and 70 kDa dextran and ficoll used in this work according to this equation is 1.8 nm, 2.3 nm and 2.8 nm, and for BSA, with a molecular weight of 66.5 kDa, 2.7 nm (see Table 1).

Table 1. Summary of the molecular weight (MW), radius r (according to Eq 4) and diffusion coefficient D (according to Eq 2, with $\eta_{av} = (3.65 \pm 0.60)$ cP) of the molecules used in this work, and three eye drugs [24]. DLS is dynamic light scattering which measures translational diffusion, as does FRAP. Time-resolved phosphorescence anisotropy measures rotational diffusion [24]. $r_{lit}$ is the literature value of the radius of the macromolecule, from the reference given.

By combining Eqs (2), (3) and (4), we obtain the following equation:

$$\tau_{1/2} = \frac{\gamma w^2 6\pi\eta}{4kT}0.67(MW)^{1/3} \qquad (5)$$

**Table 1. Summary of molecule properties used in this study, and three eye drugs.**

| | MW / kDa | r / nm | D / m s$^{-2}$ | r$_{lit}$ / nm with reference and method of measurement |
|---|---|---|---|---|
| Ficoll/dextran | 20 | 1.8 | $(3.27 \pm 0.54) \times 10^{-11}$ | |
| Ficoll/dextran | 40 | 2.3 | $(2.56 \pm 0.42) \times 10^{-11}$ | |
| BSA | 66.5 | 2.7 | $(2.18 \pm 0.36) \times 10^{-11}$ | 5.4 ± 0.1 [31] (DLS) |
| | | | | 4.8 [32] (DLS) |
| | | | | 3.49 ± 0.03 [24] (time-resolved phosphorescence anisotropy) |
| | | | | 3.29 ± 0.15 [26] (FRAP) |
| Ficoll/dextran | 70 | 2.8 | $(2.10 \pm 0.35) \times 10^{-11}$ | 3.39 [26] (FRAP) |
| | | | | 5.35 [26] (FRAP) |
| ranibizumab | 48 | 2.4 | | 4.2 [31] (DLS) |
| | | | | 4.1 [32] (DLS) |
| | | | | 2.75 ± 0.04 [24] (time-resolved phosphorescence anisotropy) |
| aflibercept | 115 | 3.2 | | 3.7 ± 0.03 [24] (time-resolved phosphorescence anisotropy) |
| bevacizumab | 150 | 3.5 | | 6.3 ± 0.1 [31] (DLS) |
| | | | | 6.5 [32] (DLS) |
| | | | | 4.58 ± 0.01 [24] (time-resolved phosphorescence anisotropy) |

Thus, the diffusion coefficient in Eq (2) can be expressed as:

$$D = \frac{kT}{6\pi\eta}\left(\frac{1}{0.67^3 MW}\right)^{1/3}$$

(6)

For the dextran and ficoll data, we used Eq (5) to plot the half-recovery time for each bleach spot size against the MW$^{1/3}$. Each data set (with the same bleach spot size) was fitted and the gradient 0.67 *(γw$^2$6πη/4kT)* was extracted. By knowing the gradient 0.67 *(γw$^2$6πη/4kT)* and the constants in Eq (5), one can calculate w$^2$η. Finally, we plotted w$^2$η against the different bleach spot radii w and from the gradient, we obtained the porcine vitreous viscosity η.

Images extracted from the Leica software were analysed with a line profile using Image J (Image J 1.45s, National Institute of Health, USA) across the bleach area in 10 different orientations going through the centre. The 10 diameters were then averaged and divided by 2 to produce a bleach radius w used for further calculations. The fitting for each FRAP recovery curve was done with a built-in MATLAB script. Experiments were carried out in triplicate for each of three different bleach areas of radius 15, 20 and 25 μm. Mean values are presented along their standard deviations.

It was important to select test agents to demonstrate the diffusion through vitreous and calculation of the diffusion coefficient. Ideally, spherical test reagents satisfying the parameters outlined in Eq (2) should have been chosen to demonstrate vitreous viscosity, however the globular BSA, linear dextran and conformable ficoll was chosen due to their size, shape, and surface charge and to allow comparisons with previous work which helped validated the present methodology [26].

## Zeta potential

The net surface charge of the ficoll and dextran was analysed using a Zetasizer Nano ZS (Malvern Instruments Ltd., UK) that uses laser Doppler velocimetry (LDV) to measure the zeta potential of particles in a solution. Samples were diluted in PBS with a concentration of 1.4 mg/mL to avoid multiple scattering and placed in a disposable zeta cell for measurements. A 4 mW HeNe 633 nm laser was shone through the sample to measure the velocity of the

molecules in an applied electric field of known value. The intensity of the scattered laser light was detected at an angle of 173˚ by an avalanche photodiode. Measurements were repeated three times and a standard deviation calculated.

## Results

Experiments were first conducted using FITC alone, but we found that FITC alone diffused too quickly based on Meyvis *et al*'s observation that the bleach time:diffusion time ratio should be more than 1:15 [33]. Therefore, fluorescently labelled BSA, ficoll and dextran were used instead. This also provided more direct comparison with earlier experiments [34]. Typical FRAP recovery curves for FITC-labelled dextran and ficoll are shown in Figs 1 and 2, where the data points prior to the first 8 seconds corresponds to the pre-bleach intensity values. The bleach occurs approximately after 9 seconds and from there the recovery of the FRAP curve takes place. These recovery curves were fitted with a bi-exponential model, (Eq (1)) to obtain the half-recovery time ($\tau_{1/2}$) for different bleach areas, and for different molecular weights. The residuals are shown below the fitted recovery curve. The residuals for ficoll in Fig 2 extend over a larger range than for dextran in Fig 1, because the ficoll recovery curve is noisier than the dextran recovery curve. The $\tau_{1/2}$ was extracted by finding the time at which the height of the FRAP recovery curve was half of its maximum value. They increased with both the molecular weight and with the bleach spot radius, as expected from Eq 5. In the examples shown in Figs 1 and 2, the diffusion coefficient was calculated from the standard FRAP Eq 3, without the use of other equations, solely from the bleach area and recovery time. They are 1.7 x $10^{-11}$ m$^2$/s for the dextran sample, and 1.5 x $10^{-11}$ m$^2$/s for the ficoll sample.

The recovery curves were fitted with a bi-exponential model, Eq 1, and the residuals are shown below the recovery curve. The recovery times $\tau_{1/2}$ from three different recovery curves were averaged. The first 8 seconds represent the pre-bleach average fluorescence in the bleach area and correspond to constant *a* in Eq 1. The half-recovery time for dextran (Fig 1) is 2.80 s, whereas for ficoll (Fig 2) is 3.26 s. The corresponding diffusion coefficients as calculated from Eq 3 is 1.7 x $10^{-11}$ m$^2$/s for the dextran sample and is 1.5 x $10^{-11}$ m$^2$/s for the ficoll sample.

For each of the seven samples, experiments were repeated six times, on the same sample, for each bleach area. Underestimated values for the diffusion coefficient *D* are often obtained

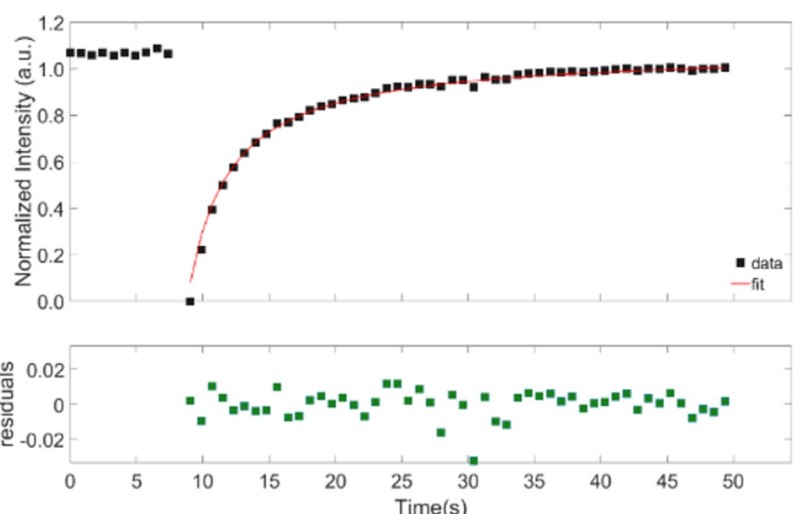

**Fig 1. Representative FRAP curves in vitreous for 70 kDa dextran with a bleach spot radius of 15 μm.**

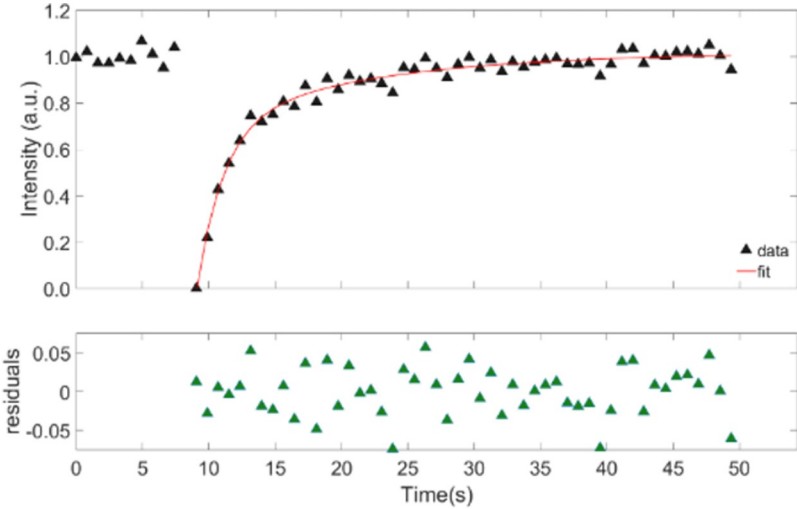

**Fig 2. Representative FRAP curves in vitreous for 70 kDa ficoll with a bleach spot radius of 15 μm.**

and can vary depending on bleaching spot sizes [35, 36], as a result of long scanning times of the confocal laser scanning microscope when viewing molecular diffusion [37]; this was mini-mised by using the raw bleach images to obtain the bleach radius. As shown in Figs 3 and 4, the recovery time of both molecules showed a linear increase with the systematically increased molecular weight to the power of 1/3, as expected from Eq (5) [35, 36].

The goodness of fit $R^2$ for dextran and the bleach radii w1 = 25 μm, w2 = 20 μm and w3 = 15 μm are 0.83, 0.94 and 0.88, respectively. For ficoll, $R^2$ per bleach radius in the same order is given by 0.86, 0.69 and 0.63 (Fig 3).

A plot of the recovery times for dextrans for different bleach areas according to Eq (5) is shown in Fig 3, and for ficolls in Fig 4. A straight line fit thorough the origin as demanded by the theoretical model, Eq (5), yields gradients 0.67 *($\gamma w^2 6\pi\eta/4kT$)* that depend on the bleach area.

For ficolls, as a rule of thumb, better quality data seems to correlate with larger bleach spot radii, where the fit for $w_1$ = 25 μm presents the highest $R^2$ value (0.86) (Fig 4). For dextrans

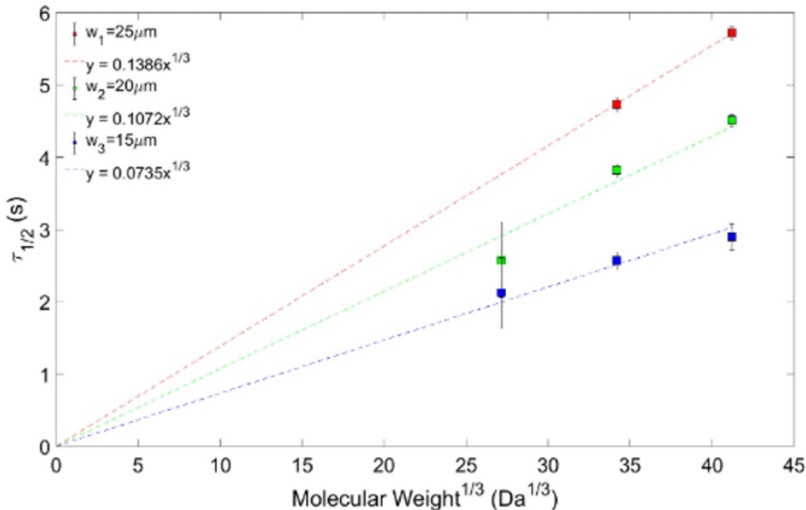

**Fig 3. Half-recovery time against MW$^{1/3}$ for dextran for different bleach radii.**

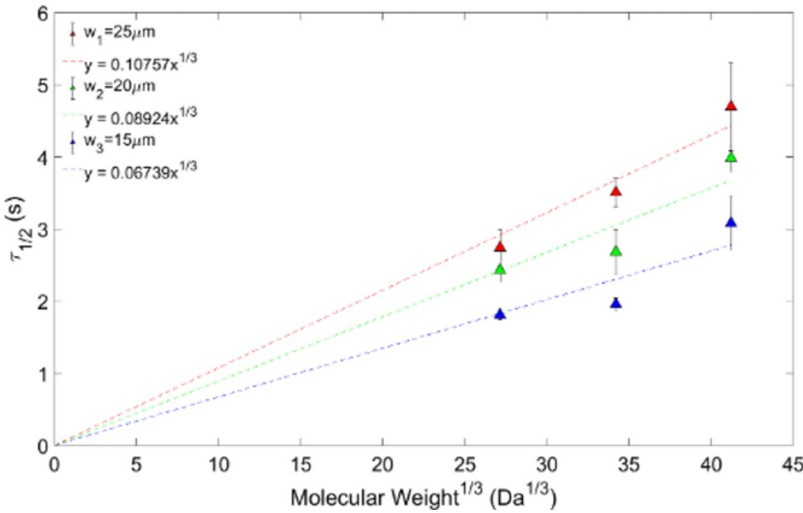

**Fig 4. Half-recovery time against MW$^{1/3}$ for ficoll for different bleach radii.**

this also seems to be the case: the 20 μm bleach radius has a larger $R^2$ value ($R^2 = 0.94$) than the 15 μm bleach radius ($R^2 = 0.88$).

Plotting the gradients $0.67(\gamma w^2 6\pi\eta/4kT))$ obtained from the straight-line fits in Figs 3 and 4 versus the bleach area also yields straight lines through the origin, as expected from Eq (5) (Figs 5 and 6).

The viscosity of the vitreous as derived from a straight line fit according to Eq 5 for a) is $3.54 \pm 1.40$ cP, and for b) $2.86 \pm 1.13$ cP. The goodness of fit $R^2$ were 0.7025 and -0.091, respectively.

The analysis of these gradients allows a self-consistent calculation of the viscosity of the vitreous, under the assumption that the dextrans and ficolls have a hard spherical shape.

Using Eq (5), we calculated the porcine vitreous viscosity from the dextran and ficoll data. For dextran, the gradients $0.67$ $(\gamma w^2 6\pi\eta/4kT)$ from the fitted data plotted on Fig 3, for the bleach spot radii of 15, 20 and 25 μm, are: $0.1386 \pm 0.0030$, $0.1072 \pm 0.0190$ and $0.0735 \pm 0.0095$ s(Da)$^{1/3}$,

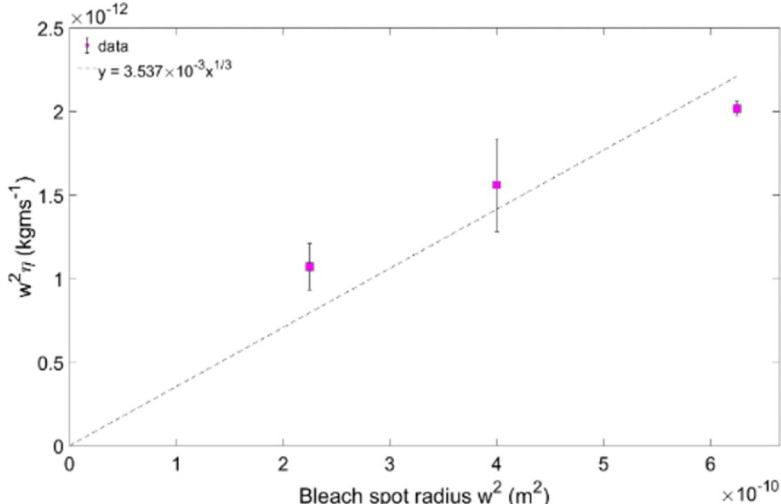

**Fig 5. Constant from Fig 3 gradients against the bleach spot radius w$^2$, for dextran.**

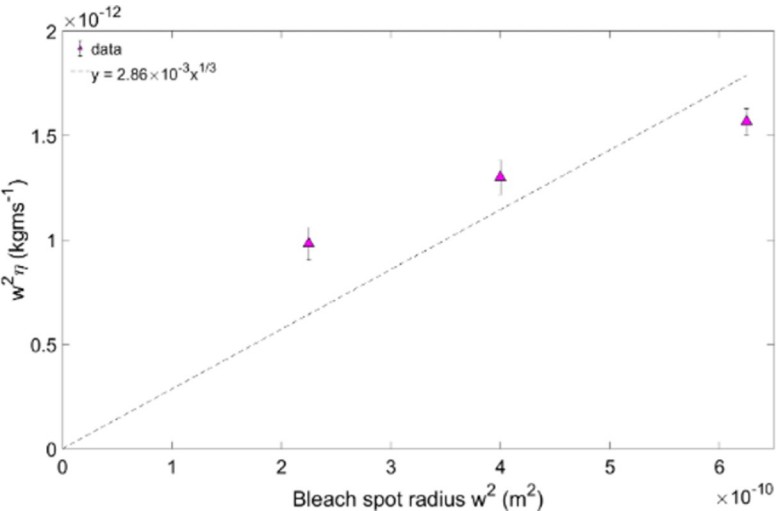

**Fig 6. Constant from Fig 4 gradients against the bleach spot radius $w^2$, for ficoll.**

respectively. In Fig 5, the viscosity is given by the gradient 0.67 *($\gamma w^2 6\pi\eta/4kT$)*, which is $\eta$ = (3.54 ± 1.40) cP. In the case of the ficoll data, the results are shown in Figs 4 and 6. In Fig 4, the gradients from the fits are: (0.1076 ± 0.0042) s(Da)$^{1/3}$, (0.0892 ± 0.0056) s(Da)$^{1/3}$ and (0.0674 ± 0.0054) s(Da)$^{1/3}$. This corresponds to a bleach spot radii of 15, 20 and 25 μm, respectively. Fig 6 gradient yields a porcine vitreous viscosity value of (2.86 ± 1.13) cP. As the vitreous viscosity value calculated from the ficoll measurements agrees with the one calculated from the dextran measurements within the experimental uncertainty, it can be said that the vitreous viscosity value is reproducible when measured using dextran.

Using these viscosity values for 70 kDa dextran and ficoll, with a radius of 2.8 nm (according to Eq 4), and employing Eq 2 to calculate a corresponding diffusion coefficient, results in a value of D = (2.2±0.9) x 10$^{-11}$ m$^2$/s for dextrans, and D = (2.7±1.0) x 10$^{-11}$ m$^2$/s for ficolls. These values are in agreement with the values directly calculated from the experimental parameters' recovery time and bleach area, and Eq 3, in Figs 1 and 2, within the experimental uncertainty.

In addition, FRAP was performed on BSA at a bleach radius of 15 μm in vitreous humor to allow direct comparison of previous experiments done in the same conditions on aqueous buffer solutions [26]. We found that the mean translational diffusion coefficient of the 66.5 kDa BSA in porcine vitreous was 14.37 ± 0.09 μm$^2$s$^{-1}$. Previously, we used the same FRAP methodology to calculate the translational diffusion coefficient of BSA in aqueous buffer solution as 6.53 ± 0.03 x 10$^{-7}$ cm$^2$/s, equivalent to 65.3 ± 0.30 μm$^2$s$^{-1}$ [26]. Therefore, porcine vitreous slows down the diffusion of BSA by a factor 4.54. Assuming a viscosity of 1 cP for the aqueous buffer solution [38], according to Eq (2) the viscosity of porcine vitreous is $\eta_{BSA}$ = (4.54 ± 0.13) cP. This value agrees well within experimental uncertainty with the vitreous viscosity calculated from the dextran FRAP experiments, $\eta_{dextran}$ = (3.54 ± 1.40) cP, and is close to the value from ficoll FRAP experiments, $\eta_{ficoll}$ = (2.86 ± 1.13) cP. An average viscosity of porcine vitreous, $\eta_{av}$, can be calculated from $\eta_{av} = (\eta_{ficoll} + \eta_{dextran} + \eta_{BSA})/3$, which yields (3.65 ± 0.60) cP.

## Discussion

Vitreous humor varies across different species. Lee *et al*. compared human and porcine vitreous [12]. They used macroscopic samples and a microrheometer to determine the viscosity.

Their studies showed that some regions of the vitreous in a pig's eye appeared to have a "thicker" gel consistency whereas human vitreous was more uniform and "thinner", even "watery", in the anterior and central portion. Samples obtained from the pig central region also presented a "thinner" consistency in the majority of cases. The range of reported viscosity therefore reflects this diversity in viscosity in different parts of the vitreous. The authors concluded that the central porcine vitreous, as sampled in the current study, was the best match to human vitreous.

Another study, by Gisladottir et al, reported that the mean (± standard deviation, SD) viscosity of porcine vitreous was 6.3 ± 2.3 cP [39]. The study used diffusion cells with a middle chamber containing porcine vitreous to determine diffusion coefficients of (non-fluorescent) macromolecules and used the Stokes-Debye-Einstein Eq 2, to calculate viscosity. They also carried out similar experiments in saline and concluded that molecules diffuse 6 times slower in vitreous, which is in agreement with our results of 4.54 times slower using FRAP on BSA molecules. The values obtained from FRAP of dextrans (3.54 ± 1.40) cP, and ficoll (2.86 ± 1.13) cP are also in good agreement, and well below the 200,000 cP for low shear stress (below 0.2 Pa) quoted for mechanical bulk measurements [3].

In addition to these two studies focussing on the viscosity of the vitreous humor, there are a number of studies investigating the macroscopic viscoelastic properties of the vitreous humor, for example its response to a mechanical stimulus, e.g. with magnetic resonance imaging [40], or ultrasound techniques [41, 42] or laser light scattering [43] on living human eyes. Magnetic microrheology, employing 0.5 mm magnetic microspheres in cadaver vitreous, applying a magnetic field and tracking their movement has also been used [44], and rheology experiments using novel cylindrical geometries of bovine and porcine vitreous humor has also been reported [45], as well as rheological creep tests to study the effect of enzymatic activity on the vitreous humor structure and its mechanical characteristics over time [46].

Vitreous rheological properties have been extensively studied with mechanical methods via the vitreous storage or elastic modulus G' and the loss or viscous modulus G" in bovine vitreous [47–49], porcine vitreous [48, 49] and human vitreous [11]. Elastic modulus G describes an object's tendency to deform its shape at constant volume when acted upon by opposing forces. It is defined as shear stress over shear strain and can be used to derive viscosity. These studies demonstrate that the vitreous elastic modulus G' is larger than the viscous modulus G", which means that the bulk vitreous behaves more like a solid than a fluid. As these studies model the bulk vitreous in its entirety, it does not reflect the differences in viscosity seen in different portions of the vitreous humor. Sharif-Kashani et al [50] showed different stress responses in vitreous between the two main biopolymer regions. They describe a solid state composed mainly of collagen fibres and a more liquid state of hyaluronan network and dispersed microfibrils.

Our study used FRAP to directly measure macromolecular diffusion in vitreous on a micrometer scale. We make the assumption that the diffusing molecules behave as nanometer-sized spherical objects. As the radius of a molecule increases, the MW is predicted to increase by a cubed factor of the radius, if we assume a spherical model. Dextrans have been well characterized [51, 52] and it seems likely that they exist as a branching dendrimer or random coil [53]. However, Burchard found that as their molar mass increases, the hydrodynamic properties of dextrans approach that of compact hard spheres due to an increase in branching density [54]. In our previous experiments we found that in solution, ficoll behaves very similarly to BSA [26]. Bovine serum albumin is known to exist as a compact sphere [26] and this result suggests ficoll does likewise in solution, even though it can assume a more open configuration [26, 55]. The assumption that the diffusing molecules behave as spherical objects is therefore justified, and in agreement with the assumptions made in deriving the equations used to

analyse the data, e.g. the Stokes Einstein Eq 2, and the hard sphere model for proteins, Eq 4. Time-resolved phosphorescence anisotropy measurements [24] could be used to try and elucidate deviations from the assumption of a spherical shape. They are more sensitive to the molecular shape than FRAP, as they depended on the cube on the molecular radius, rather the radius as in FRAP, but we did not perform these measurements here.

We find that increasing MW increases the half-recovery time. This is qualitatively expected, as due to the larger MW, the diffusion is slowed, and quantitatively follows Eq 5, which shows that the recovery time is proportional to the cubed root of the molecular volume. Increasing the bleach area in the FRAP experiments also increases the recovery time. This can be rationalised due to the fact that a larger area takes longer to repopulate with unbleached fluorophores, which is quantitatively also contained in Eq 5. The diffusion constant in Eq 3 is a feature of the system under study, independent of the instrumental settings to measure it, and hence the ratio of bleach area and recovery time remains constant–if one increases, so does the other. The systematic variation of these two independent parameters, MW and bleach area, thus allows the extraction of the viscosity of the vitreous from the theoretical framework.

The nanometer-sized molecules report on the viscosity experienced by their environment, on the micrometer scale. This is similar to the scenario in the work reported by Tuteja *et al.* who measured the diffusion coefficients of sub-10 nm diameter nanoparticles in polymer liquid using X-ray photon correlation spectroscopy [56]. They proposed that when a molecule is extremely small (sub-10 nm diameter), it most likely diffuses through the vitreous humor in the canals between the collagen polymer chains, and therefore the macroscopic viscosity of the vitreous gel might not accurately predict diffusion on the nanoscale in such a heterogeneous medium.

Neutral charge molecules were selected for this study, but despite this the zeta potentials of 'neutral' ficoll and dextran were–(4.79 ± 2.05) mV and–(2.68 ± 0.56) mV respectively. Hyaluronan is the most common vitreous glycosaminoglycan and holds a negative charge. It is possible that the small negative charge of dextran and ficoll speeds diffusion, due to repulsion from the negatively charged vitreous, as is thought to occur in other ocular tissue [57, 58]. Dextran diffusion was slower than albumin in solution, but it was very similar in vitreous. It could be that dextran may assume a more compact, spherical shape in vitreous. Alternatively, or perhaps as well, it may be that the negative dextran charge speeds diffusion.

Other studies also suggest that surface charge may influence vitreous diffusion. Ruponen *et al.* found that negatively charged hyalonan interacts with polymeric and liposomal DNA complexes [59]. Previous in-vitro studies have shown that cationic liposome complexes aggregate in the vitreous [60]. By increasing the degree of pegylation of the liposome and decreasing the zeta potential to become anionic, the authors observed homogeneous spreading of non-aggregated anionic liposome. Similarly, Kim *et al.* found that intravitreally administered anionic human serum albumin nanoparticles diffused freely in the posterior direction from the vitreous to the retina whereas intravitreally administered cationic nanoparticles were bound and aggregated to the vitreous [61].

Experiments were conducted the day after eyes were harvested. The vitreous collapses rapidly post mortem and eyes should ideally be used within 2h of animal slaughter [46], and in addition, we have a delay before preparation, and the incubation in the FITC-labelled macromolecule solution (neither more than 24 h). It is possible that post-mortem changes alter the measured viscosity value of vitreous. However, we anticipate that post mortem change might have less impact on our comparisons between molecules, and our assessment of the utility of FRAP in vitreous. These are limitations of our study, which may contribute to the large standard deviation, but, nevertheless, the results we obtain with our optical method are in good agreement with the study by Gisladottir et al [39], who also specifically study the viscosity of vitreous humor, via diffusion, rather than its viscoelastic properties.

While our data cannot be used to estimate the effect of age, or topographical variations in viscosity within the vitreous—since we only sampled the central vitreous—future studies might consider sampling from a range of vitreous locations.

Our study used physical diffusion experiments to demonstrate the movement of molecules though vitreous. This provides a more sensitive and accurate method of calculating viscosity relevant for drug diffusion as values used for calculations that are obtained from direct mechanical bulk measurements. Previous studies relied on microrheometry such as those carried out by *Lee et al.* [11, 12, 19]. Although, since the early 1990s when these experiments were performed, microrheometry has advanced significantly, direct diffusion experiments employing small macromolecules similar in MW to drugs may provide a more accurate and sensitive approach to viscosity measurements, for both real vitreous and vitreous substitutes [25].

In summary, this study shows that optical microviscosity measurements as performed by FRAP, using labelled molecules with a MW similar to those of intravitreal drugs, may provide a useful means with which to directly measure the intravitreal diffusion. Using a bespoke theoretical framework to analyse the data, we calculate a porcine vitreous viscosity of $(3.65 \pm 0.60)$ cP. FRAP may help quantify the diffusion of ocular macromolecules in health and disease and may help predict the pharmacokinetics of intravitreal drugs. This may be relevant to the intravitreal pharmokinetics of drugs such as bevacizumab (150 kDa), ranibizumab (48 kDa) and aflibercept (115 kDa) [21–24]. While the MW of ranibizumab (48 kDa) falls within the range of MW of the macromolecules in this study (20 kDa, 40 kDa and 70 kDa), the diffusion coefficients of the larger bevacizumab (150 kDa) and aflibercept (115 kDa) can extrapolated from Eq 2, as summarised in Table 1. Thus FRAP of fluorescently labelled macromolecules is a feasible and practical optical method to quantify their diffusion through vitreous.

## Acknowledgments

The authors thank Dr James Levitt and Pei-Hua Chung in the Department of Physics at King's College London for their expertise in FRAP.

## Author Contributions

**Conceptualization:** Nishanthan Srikantha.

**Data curation:** Nishanthan Srikantha, Andrew Simpson, Naba Elsaid.

**Formal analysis:** Nishanthan Srikantha, Yurema Teijeiro-Gonzalez.

**Investigation:** Nishanthan Srikantha, Naba Elsaid.

**Methodology:** Nishanthan Srikantha.

**Supervision:** Satyanarayana Somavarapu, Klaus Suhling, Timothy L. Jackson.

**Writing – original draft:** Nishanthan Srikantha.

**Writing – review & editing:** Yurema Teijeiro-Gonzalez.

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
