## [Decision Letter · Decision Letter 0]

2 Jul 2021

PONE-D-21-17834

Determining Vitreous Viscosity using Fluorescence Recovery after Photobleaching (FRAP)

PLOS ONE

Dear Dr. Srikantha,

Thank you for submitting your manuscript to PLOS ONE. After careful consideration, we feel that it has merit but does not fully meet PLOS ONE’s publication criteria as it currently stands. Therefore, we invite you to submit a revised version of the manuscript that addresses the points raised during the review process.

We look forward to receiving your revised manuscript.

Kind regards,

Paula Schaiquevich

Academic Editor

PLOS ONE

Additional Editor Comments (if provided):

The manuscript entitled " Determining Vitreous Viscosity using Fluorescence Recovery after Photobleaching (FRAP)" by Nishanthan Srikantha et al presents novel data on the analysis of the viscosity of vitreous humor. This topic has clinical implications and in my opinion, contributes to the basic understanding but also, has potential clinical translation.

There are some aspects that should be addressed that will greatly improve the manuscript and are detailed by the reviewers.

Yours sincerely,

Paula Schaiquevich

Journal Requirements:

"The project was funded by Fight for Sight Grant Number 1874. N. Srikantha is a Fight for Sight Research Fellow."

"Supported by Fight for Sight Grant Number 1874. Nishanthan Srikantha is a Fight for Sight Fellow. The authors thank Dr James Levitt and Pei-Hua Chung in the Department of Physics at King’s College London for their expertise in FRAP. The author and co-authors do not have any conflict of interests."

"The project was funded by Fight for Sight Grant Number 1874. N. Srikantha is a Fight for Sight Research Fellow."

Reviewers' comments:

Reviewer's Responses to Questions

**Comments to the Author**

1. Is the manuscript technically sound, and do the data support the conclusions?

Reviewer #1: Partly

Reviewer #2: Partly

2. Has the statistical analysis been performed appropriately and rigorously? 

Reviewer #1: No

Reviewer #2: No

3. Have the authors made all data underlying the findings in their manuscript fully available?

Reviewer #1: Yes

Reviewer #2: Yes

4. Is the manuscript presented in an intelligible fashion and written in standard English?

Reviewer #1: Yes

Reviewer #2: Yes

5. Review Comments to the Author

Reviewer #1: The authors presented a FRAP (Fluorescence Recovery After Photobleaching) method to study intravitreal diffusion and measured viscosity of the vitreous humor.

This study is significant because of the microscopy technique limitations for molecules below 50 nm. For example, it is difficult to study a 30nm size particle ex vivo or an Ab molecule. This paper sets an interesting direction in molecular diffusion study.

Some concerns are listed below:

Materials section:

121-125 it is a bit confusing. Modify it in a simple manner. What do you mean by central vitreous and central portion? How much vitreous amount and what is amount and volume of soaked molecules. Clearly mention concentrations if any.

What is the molar ratio of fluorophore to proteins? mention in manuscript.

161 : Equation: define every parameter and units clearly.

166-168 line and table 1 : What is DLS or AF4 data for BSA molecule and other molecules present in the table.

It seems the estimates are relatively small in number why is that? If you have literature values for radius or diameter refer it in the table.

200-203 lines 2.4 You measured zeta potential values. Present them in results. It is possible that the buffer you used might influence the charge so please describe what is the measuring medium? What was the electrophoretic mobility and conductivity values in Zeta sizer? How many measurements were used?

Did you measure molecule size in zeta sizer? Present the data from experiment or from literature.

206-207 Where is the data presented for ANOVA?

Results section:

Add repeats in all figure legends. Also clearly mention which molecular weights in legends for all figures.

Why did you exclude other molecules from table 1? Justify in discussion section.

Improve the resolution of figures.

Discussion:

There are interesting papers published after 2018. Update the references and discuss the same in introduction and discussion section.

446 what is the justification for this statement? PK study is more complex in nature with half lives in days to month.

49 line This conclusion is an extrapolation, the data presented this paper (the study duration) is not enough to study the pharmacokinetics. However, it will show vitreal interactions and retention behavior. Remove PK or justify how? with duration.

Other comments:

Instead of using dissected vitreous is it possible to do this experiment on sectioned eye? How you ensure that the vitreous structure is not disturbed?

What are the limitations or advantages of using half cut eye model described in the latest papers?

Reviewer #2: The manuscript " Determining Vitreous Viscosity using Fluorescence Recovery after Photobleaching (FRAP)” from Nishanthan Srikantha et al., addresses a significant topic both from the clinical and technological point of view. Aim of the present study is to evaluate the viscosity of vitreous humor via diffusion, thus avoiding the variability of the positioning of the chains of the gel-like structure during mechanical stresses.

Although the experiments were carried out with reliable and reproducible methods, the Authors did not justify some procedural choices.

Please find below major concerns and suggestions:

- The vitreous presents spatial and temporal variations of its rheological properties from individual to individual, as well as eyes from the same owner. The number of pig eyes used for the present work is very low for representing different ocular conditions, considering also that only the central portions of the vitreous were taken (choice justified by the Authors).

- The role of temperature is fundamental for evaluating sample’s viscosity. The method section does not provide any information about the temperature at which FRAP experiments have been performed. The reader expects the experiments to be conducted under conditions close to physiological ones.

- As stated in the manuscript (lines 417-419), vitreous experiences changes in macromolecular organization after dissection (A. F. da Silva, M. A. M. Alves, and M. Oliveira. Rheological behaviour of vitreous humour. Rheologica Acta, 56:377–386, 2017). Experiments reported in the present work have been conducted after 24 hours of eyes storage. Excluding the 24 h for FITC labelling, it is of significant relevance to evaluate viscosity and diffusion results for different dissection time.

For the above reasons, I believe that this work could be deeply improved with a wider experimental campaign which takes into account the variability of conditions mentioned before.

Minor revisions:

-line 96: MW to be defined, since mentioned for the first time.

-line 225: as calculated -> was calculated

-lines 257-259: spot radii in mm -> μm

6. PLOS authors have the option to publish the peer review history of their article (what does this mean?). If published, this will include your full peer review and any attached files.

Reviewer #1: No

Reviewer #2: No

---

## [Author Response · Author response to Decision Letter 0]

12 Aug 2021

Dear Dr Schaiquevich,

we would like to thank the reviewers for the positive and constructive comments, and you for the editorial feedback.

We have addressed the comments as follows:

We have now followed the guidance in these two documents

"The project was funded by Fight for Sight Grant Number 1874. N. Srikantha is a Fight for Sight Research Fellow."

We have now deleted this statement in the manuscript because you say in 3) below that “funding information should not appear in the acknowledgements section of in other areas of your manuscript.”

"Supported by Fight for Sight Grant Number 1874. Nishanthan Srikantha is a Fight for Sight Fellow. The authors thank Dr James Levitt and Pei-Hua Chung in the Department of Physics at King’s College London for their expertise in FRAP. The author and co-authors do not have any conflict of interests."

"The project was funded by Fight for Sight Grant Number 1874. N. Srikantha is a Fight for Sight Research Fellow."

We have removed the funding statement from the Acknowledgements section, so that it now reads:

“The authors thank Dr James Levitt and Pei-Hua Chung in the Department of Physics at King’s College London for their expertise in FRAP.”

We would like the Funding Statement to read as follows:

The project was funded by Fight for Sight Grant Number 1874. The funders had no role in study design, data collection and analysis, decision to publish, or preparation of the manuscript. N. Srikantha is a Fight for Sight Research Fellow. K. Suhling acknowledges BBSRC grant BB/R004803/1. 

5. Review Comments to the Author

Reviewer #1: The authors presented a FRAP (Fluorescence Recovery After Photobleaching) method to study intravitreal diffusion and measured viscosity of the vitreous humor.

This study is significant because of the microscopy technique limitations for molecules below 50 nm. For example, it is difficult to study a 30nm size particle ex vivo or an Ab molecule. This paper sets an interesting direction in molecular diffusion study.

Some concerns are listed below:

Materials section:

121-125 it is a bit confusing. Modify it in a simple manner. What do you mean by central vitreous and central portion? 

Apologies for the sloppy terminology. We mean the vitreous from the centre of the eye. It had less than 1 cm diameter. We have now replaced

“the central portion was carefully removed” 

with 

“the central region with a diameter less than 1 cm was carefully removed”

How much vitreous amount and what is amount and volume of soaked molecules. Clearly mention concentrations if any.

Apologies, this was unclear. We used about 1 mL of vitreous, and soaked it in about 5 mL of 10-5 M solution with labelled ficoll or dextran. Instead of 

“These three separate central vitreous body samples were soaked for 24 hours in fluorescein-isothiocyanate (FITC) labelled, neutral dextran or ficoll…..”

We now state 

“These three separate central vitreous body samples of around 1 mL volume each were soaked for 24 hours in about 5 mL of 10-5 molar fluorescein-isothiocyanate (FITC) labelled, neutral dextran or ficoll….”

What is the molar ratio of fluorophore to proteins? mention in manuscript.

According to the manufacturer, TdB, “The fluorescein moiety is attached by a stable thiocarbamoyl linkage and the labelling procedure does not lead to any depolymerization of the dextran. FITC-dextrans have from 0.002-0.008 mol FITC per glucose unit. These low levels of substitution confer minimal charges to dextran, which is an essential requirement for permeability studies.”

https://tdblabs.se/products/fluorescent-derivatives/fitc-derivatives/fitc-dextran/

The manufacturer also states

Polysucrose (renamed from Ficoll™) is a polymer which displays excellent biocompatibility. Polysucrose is a polymer synthesised by cross-linking sucrose with epichlorohydrin. FITC-Polysucrose is prepared by reacting polysucrose with FITC under similar conditions to those used for FITC-dextrans. Polysucrose has a more spherical structure and is less flexible compared to dextran.

https://tdblabs.se/products/fluorescent-derivatives/fitc-derivatives/fitc-polysucrose/

We now state in the manuscript:

“According to the manufacturer, the fluorescein moiety is attached by a stable thiocarbamoyl linkage and the labelling procedure does not lead to any depolymerization of the dextran. FITC-dextrans have from 0.002-0.008 mol FITC per glucose unit. Ficoll (polysucrose) is a polymer synthesised by cross-linking sucrose with epichlorohydrin. FITC-ficoll is prepared by reacting ficoll with FITC under similar conditions to those used for FITC-dextrans. Ficoll has a more spherical structure and is less flexible compared to dextran.”

161 : Equation: define every parameter and units clearly.

Apologies, we omitted the half-recovery time τ1/2, we have now included it.

We are not sure what the reviewer means by “define units clearly”. The equation is valid for any units as long as the parameters have the right dimensions. For example, the dimension of time can be measured in units of seconds or hours, and the dimension of radius is a length which can be measured in inches or centimeters or light years. For clarity, we have added

“The diffusion coefficient has dimensions of [length]2/[time] = [area] / [time].”

166-168 line and table 1 : What is DLS or AF4 data for BSA molecule and other molecules present in the table. It seems the estimates are relatively small in number why is that? If you have literature values for radius or diameter refer it in the table.

The values in the table are calculated according to eq 2, and give a minimum value for the radius assuming the molecule is spherical, according to

Erickson HP. Size and shape of protein molecules at the nanometer level determined by sedimentation, gel filtration, and electron microscopy. Biol Proced Online. 2009;11:32–51.

We have now cited this reference, and added results from DLS, FRAP and time-resolved phosphorescence anisotropy measurements to the table. 

200-203 lines 2.4 You measured zeta potential values. Present them in results. 

They are stated and discussed in paragraph 10 of the discussion section. We have now added the standard deviation to the values.

It is possible that the buffer you used might influence the charge so please describe what is the measuring medium? What was the electrophoretic mobility and conductivity values in Zeta sizer? How many measurements were used?

The measuring medium is PBS buffer, and measurements were repeated three times. We now state this explicitly in the Methods section:

“Samples were diluted in PBS with a concentration of 1.4 mg/ml to avoid multiple scattering and placed in a disposable zeta cell for measurements. A 4-mW He-Ne 633 nm laser was shone through the sample to measure the velocity of the molecules in an applied electric field of known value. The intensity of the scattered laser light was detected at an angle of 173° by an avalanche photodiode. Measurements were repeated three times and a standard deviation calculated.”

Did you measure molecule size in zeta sizer? Present the data from experiment or from literature.

No, we did not measure the molecular size in the zeta sizer.

206-207 Where is the data presented for ANOVA?

We have now deleted the reference to ANOVA. 

Results section:

Add repeats in all figure legends. Also clearly mention which molecular weights in legends for all figures.

We have changed the previous caption

“Fig. 1. Example FRAP curves in vitreous for 70 kDa dextran (A), and ficoll (B) with a bleach spot radius of 15 μm. The recovery curves were averaged and fitted with a bi-exponential model, equation 1.”

to now read

“Fig. 1. Representative FRAP curves in vitreous for 70 kDa dextran (A), and 70 kDA ficoll (B) with a bleach spot radius of 15 μm. The recovery curves were fitted with a bi-exponential model, equation 1, and the residuals are shown below the recovery curve. The recovery times τ1/2 from three different recovery curves were averaged.”

Why did you exclude other molecules from table 1? Justify in discussion section.

We have now added bevacizumab (150 kDa), ranibizumab (48 kDa) and aflibercept (115 kDa) to the table, and included literature radii, references and the method of measurement in a new column the table.

Improve the resolution of figures.

Discussion:

There are interesting papers published after 2018. Update the references and discuss the same in introduction and discussion section.

We agree, many interesting papers have been published since 2018 regarding optical diffusion measurements, the viscosity of the vitreous humor and work on eye drugs. 

However, a specific search of pubmed (fluorescence recovery after photobleaching or FRAP) AND vitreous does not reveal any other recent papers than the ones cited already, but we would be happy to include any advocated by the reviewer.

Two particular topics caught our eye that we decided to elaborate on: MRI measurements of eyes and substitute materials for the vitreous humor.

MRI measurement of porcine eyes are performed in these interesting publications:

Stein S, Hadlich S, Langner S, Biesenack A, Zehm N, Kruschke S, Oelze M, Grimm M, Mahnhardt S, Weitschies W, Seidlitz A (2018): 7.1 T MRI and T2 mapping of the human and porcine vitreous body post mortem, European Journal of Pharmaceutics and Biopharmaceutics 131, 82-91.

Thakur SS, Pan X, Kumarasinghe GL, Yin N, Pontré BP, Vaghefi E, Rupenthal ID (2020): Relationship between rheological properties and transverse relaxation time (T2) of artificial and porcine vitreous humour, Experimental Eye Research 194, 108006.

We have added a brief mention of this topic in the introduction:

“Magnetic resonance imaging (MRI) has also been employed not only to measure, but also to image the rheological properties of vitreous, both in porcine [60] and human eyes (post mortem). [61] The so-called transverse relaxation time T2 is correlated to viscoelastic properties of the vitreous, i.e. the storage or loss moduli (which are related to the damping properties of the material) and the viscosity. The great advantage of this approach is that it maps T2 and thus provides images which allows spatial variation of viscoelastic properties to be visualised. For example, the central vitreous has a slightly shorter T2 compared to the peripheral vitreous. [60, 61]”

Regarding vitreous substitute materials, they should have very similar properties to the original, as detailed in this interesting recent paper:

Schulz A, Januschowski K, Szurman P (2021): Novel vitreous substitutes: the next frontier in vitreoretinal surgery, Current Opinion in Ophthalmology: 32(3): 288-293

We have added

“Moreover, polymeric hydrogels have been investigated as vitreous substitutes. [59] In addition to biocompatibility and functionality, the optical and rheological properties of these substitute materials, e.g. refractive index, density, water content, stiffness, viscoelasticity and also viscosity, should be as close as possible to the original vitreous humor.”

to the introduction, and at the end in the discussion, after “….may provide a more accurate and sensitive approach to viscosity measurements” we have now added:

“for both real vitreous and vitreous substitutes. [59]”

446 what is the justification for this statement? PK study is more complex in nature with half lives in days to month.

We agree, have deleted this statement and replaced it with

“Thus FRAP of fluorescently labelled macromolecules is a feasible and practical optical method to quantify their diffusion through vitreous.”

49 line This conclusion is an extrapolation, the data presented this paper (the study duration) is not enough to study the pharmacokinetics. However, it will show vitreal interactions and retention behavior. Remove PK or justify how? with duration.

We have shortened the conclusions to “FRAP is a feasible and practical optical method to quantify the diffusion of macromolecules through vitreous.”

Other comments:

Instead of using dissected vitreous is it possible to do this experiment on sectioned eye? 

Yes, this would be possible

How you ensure that the vitreous structure is not disturbed?

By handling it very carefully – ideally, we would like to do in-vivo experiments.

What are the limitations or advantages of using half cut eye model described in the latest papers?

An advantage of a half-cut eye model would be that dissection is not necessary, thus the risk of disturbing the vitreous is avoided. The limitations are that a half-eye model would still not be an in-vivo experiment, which we would consider the gold standard of experimentation. 

Reviewer #2: The manuscript " Determining Vitreous Viscosity using Fluorescence Recovery after Photobleaching (FRAP)” from Nishanthan Srikantha et al., addresses a significant topic both from the clinical and technological point of view. Aim of the present study is to evaluate the viscosity of vitreous humor via diffusion, thus avoiding the variability of the positioning of the chains of the gel-like structure during mechanical stresses.

Although the experiments were carried out with reliable and reproducible methods, the Authors did not justify some procedural choices.

Please find below major concerns and suggestions:

- The vitreous presents spatial and temporal variations of its rheological properties from individual to individual, as well as eyes from the same owner. The number of pig eyes used for the present work is very low for representing different ocular conditions, considering also that only the central portions of the vitreous were taken (choice justified by the Authors).

We agree with these statements and these assessments, and we would propose to investigate a larger number of samples, from different regions of the eye, to account for the biological variability. We now state in the discussion

“While our data cannot be used to estimate the effect of age, or topographical variations in viscosity within the vitreous - since we only sampled the central vitreous - future studies might consider sampling from a range of vitreous locations.” 

- The role of temperature is fundamental for evaluating sample’s viscosity. The method section does not provide any information about the temperature at which FRAP experiments have been performed. The reader expects the experiments to be conducted under conditions close to physiological ones.

We agree with the reviewers that temperature is of great importance for viscosity measurements. For example, the viscosity of high-viscosity solvent glycerol changes from around 1400 cP at 20 degrees centigrade to 900 cP at 25 degrees centigrade.

Fortunately, in the present experiments, the viscosity of the vitreous is mucher lower than that of glycerol, and thus the change with temperature is much less in absolute (and also relative) terms. For water, the viscosity only drops from 1 cP at 20 degrees to 0.9 cP at 25 degrees, and the viscosity of the vitreous is much closer to water than to glycerol. 

Moreover, the standard deviation of the value for the viscosity of the vitreous is rather large, and temperature is not the limiting factor. 

The experiments were performed at room temperature, and we now explicitly state this in the manuscript in the text after equation (1):

“….and the experiments were performed at room temperature.”

- As stated in the manuscript (lines 417-419), vitreous experiences changes in macromolecular organization after dissection (A. F. da Silva, M. A. M. Alves, and M. Oliveira. Rheological behaviour of vitreous humour. Rheologica Acta, 56:377–386, 2017). Experiments reported in the present work have been conducted after 24 hours of eyes storage. Excluding the 24 h for FITC labelling, it is of significant relevance to evaluate viscosity and diffusion results for different dissection time.

We agree with the reviewer, but at present we are unable to conduct further experiments. We do not believe that this diminishes the feasibility of our approach, and the insight gained.

For the above reasons, I believe that this work could be deeply improved with a wider experimental campaign which takes into account the variability of conditions mentioned before.

We fully agree with the reviewer, and if we had personnel and funding, we could study these conditions systematically. However, at this stage we believe it is still worth reporting our findings as a snapshot of our efforts to date. 

In addition, we note there are practical consideration sourcing eyes from an abbatoir that is distant from the laboratory and starting experiments the same day, such that much shorter timeframes might be difficult (albeit not impossible). It would be possible to do experiments over a longer interval but that would not necessarily replicate the in vivo situation any better. We do however acknowledge this point and have added the following to the discussion of limitations. We have changed

“The vitreous collapses rapidly post mortem and eyes should ideally be used within 2h of animal slaughter [40], and in addition, we have a delay before preparation, and the incubation in the FITC-labelled macromolecule solution (neither more than 24 h). 

These are limitations of our study, which may contribute to the large standard deviation, but, nevertheless, the results we obtain with our optical method are in good agreement with the study by Gisladottir et al [33], who also specifically study the viscosity of vitreous humor, via diffusion, rather than its viscoelastic properties.”

To now read

“Experiments were conducted the day after eyes were harvested. The vitreous collapses rapidly post mortem and eyes should ideally be used within 2h of animal slaughter [40], and in addition, we have a delay before preparation, and the incubation in the FITC-labelled macromolecule solution (neither more than 24 h). It is possible that post-mortem changes alter the measured viscosity value of vitreous. However, we anticipate that post mortem change might have less impact on our comparisons between molecules, and our assessment of the utility of FRAP in vitreous. These are limitations of our study, which may contribute to the large standard deviation, but, nevertheless, the results we obtain with our optical method are in good agreement with the study by Gisladottir et al [33], who also specifically study the viscosity of vitreous humor, via diffusion, rather than its viscoelastic properties.”

Minor revisions:

-line 96: MW to be defined, since mentioned for the first time.

Thank you – we have spelled out MW as molecular weight now

-line 225: as calculated -> was calculated

Thank you, we have amended this mistake

-lines 257-259: spot radii in mm -> μm

Apologies for this oversight, we have now also fixed this.

We hope that this manuscript is now acceptable for publication in PlosOne, and we look forward to hearing from you in due course.

Yours sincerely

Nish Srikantha

---

## [Editor Report · Decision Letter 1]

14 Dec 2021

Determining Vitreous Viscosity using Fluorescence Recovery after Photobleaching (FRAP)

PONE-D-21-17834R1

Dear Dr. Srikantha,

We’re pleased to inform you that your manuscript has been judged scientifically suitable for publication and will be formally accepted for publication once it meets all outstanding technical requirements.

Kind regards,

Amiel Yebsen Garcia Pimentel

Support Staff - Editorial

PLOS ONE

Additional Editor Comments (optional):

I went over again through the responses of the reviewers and despite I find that the authors could have done some more experiments, the manuscript could be accepted in the present format.

I think that they have addressed the reviewer's opinion and suggestions and therefore, I suggest the manuscript to be accepted for publication.
---

## [Editor Report · Acceptance letter]

21 Jan 2022

PONE-D-21-17834R1 

Determining vitreous viscosity using fluorescence recovery after photobleaching 

Dear Dr. Srikantha:

I'm pleased to inform you that your manuscript has been deemed suitable for publication in PLOS ONE. Congratulations! Your manuscript is now with our production department. 

Kind regards, 

on behalf of

Dr. Paula Schaiquevich 

Academic Editor

PLOS ONE